# A comparative analysis of 24-hour movement behaviors features using different accelerometer metrics in adults: Implications for guideline compliance and associations with cardiometabolic health

Iris Willems[1,2], Vera Verbestel [3,4], Dorothea Dumuid[5], Patrick Calders[1], Bruno Lapauw[6], Marieke De Craemer [1] *

1 Department of Rehabilitation Sciences, Ghent University, Ghent, Belgium, 2 Research Foundation Flanders, Brussels, Belgium, 3 Department of Health Promotion, Research Institute of Nutrition and Translation Research in Metabolism (NUTRIM), Maastricht University, Maastricht, The Netherlands, 4 Department of Health Promotion, Care and Public Health Research Institute (CAPHRI), Maastricht University, Maastricht, The Netherlands, 5 Alliance for Research in Exercise, Nutrition and Activity, Allied Health & Human Performance, University of South Australia, Adelaide, SA, Australia, 6 Department of Internal Medicine and Pediatrics & Department of Endocrinology, Ghent University Hospital & Ghent University, Ghent, Belgium

* Marieke.decraemer@ugent.be

**Data Availability Statement:** The configuration file of data processing in R package GGIR is available

# Abstract

## Background

Movement behavior features such as time use estimates, average acceleration and intensity gradient are crucial in understanding associations with cardiometabolic health. The aim of this study was to 1) compare movement behavior features processed by commonly used accelerometer metrics among adults (i.e. Euclidian Norm Minus One (ENMO), Mean Amplitude Deviation (MAD) and counts per minute (CPM)), 2) investigate the impact of accelerometer metrics on compliance with movement behavior guidelines, and 3) explore potential variations in the association between movement behavior features and cardiometabolic variables depending on the chosen metric.

## Methods

This cross-sectional study collected movement behavior features (Actigraph GT3X+) and cardiometabolic variables. Accelerometer data were analyzed by four metrics, i.e. ENMO, MAD, and CPM vertical axis and CPM vector magnitude (GGIR). Intraclass correlations and Bland–Altman plots identified metric differences for time use in single movement behaviors (physical activity, sedentary behavior), average acceleration and intensity gradient. Regression models across the four metrics were used to explore differences in 24-hour movement behaviors (24h-MBs; compositional variable) as for exploration of associations with cardiometabolic variables.

regarding the ENMO metrics in the Supporting Information S8. The same configuration was replicated for MAD, Counts Per Minute Vetrical Axis, Counts Per Minute Vector Magnitude. See Supporting Information S1 for different specification regarding the metrics. The dataset generated and analyzed during the current study is available in the Zenodo Repository [https://zenodo.org/records/10628841].

**Funding:** I.W. is supported by the Research Foundation Flanders (FWO-11N0422N) (https://www.fwo.be/). The funders had no role in the study design, data collection and analysis, decision to publish, or preparation of the manuscript.

**Competing interests:** The authors have declared that no competing interests exist.

## Results

Movement behavior data from 213 Belgian adults (mean age 45.8±10.8 years, 68.5% female) differed according to the metric used, with ENMO representing the most sedentary movement behavior profile and CPM vector magnitude representing the most active profile. Compliance rates for meeting integrated 24h-MBs guidelines varied from 0–25% depending on the metric used. Furthermore, the strength and direction of associations between movement behavior features and cardiometabolic variables (body mass index, waist circumference, fat% and HbA1c) differed by the choice of metric.

## Conclusion

The metric used during data processing markedly influenced cut-point dependent time use estimates and cut-point independent average acceleration and intensity gradient, impacting guideline compliance and associations with cardiometabolic variables. Consideration is necessary when comparing findings from accelerometry studies to inform public health guidelines.

## Background

Physical activity (PA), sedentary behavior (SB), and sleep are behaviors that are intrinsically part of an individual's daily routine; collectively known as 24-hour movement behaviors (24h-MBs) [1]. Studying their interrelatedness rather than considering them in isolation is associated with favorable health outcomes among adults [1]. By focusing on all behaviors conducted in one day, the concept of combined movement behavior guidelines has emerged, including recommendations to accumulate 150 to 300 minutes of moderate-to-vigorous PA (MVPA), limit SB, and obtain seven to nine hours of sleep with consistent bed and wake-up times [2, 3]. Despite the clear benefits of adhering to the guidelines, compliance is low among adults (approx. 7% of Canadian adults) and even worse in adults with chronic conditions such as obesity [1, 4, 5].

To better understand 24h-MBs in adults, it is crucial to accurately measure these behaviors by measurement tools such as tri-axial accelerometers (e.g. the Actigraph GT3X+) [6]. These measurement tools are the preferred method for collecting 24h-MBs as they quantify accelerations in orthogonal directions of a three dimensional space [6, 7]. There has been a shift from analyzing accelerometer data using "activity counts per minute" generated by closed-source proprietary accelerometer brand-specific algorithms (e.g. ActiLife software for Actigraph accelerometers) toward analyzing raw gravitational acceleration data (m/s$^2$) [8]. Raw acceleration data allows for open-source data processing, such as the R package GGIR, which can be used regardless of the type of accelerometer [8, 9]. Output from this open-source raw data package can be classified as time spent in movement behaviors defined by cut-points to classify activity intensities, as well as newer cut-point independent movement behavior features such as average acceleration and intensity distribution of activity throughout a day. These cut-point independent movement behavior features enhance comparability between studies [10, 11].

Nevertheless, working with raw data still requires the use of data reduction methods, also called metrics, to separate the acceleration signal from the gravitation signal [8, 9]. Different metrics exist and these can be distinguished based on the method for extracting the acceleration signal [7, 8]. Commonly used data reduction metrics for processing raw accelerometer

data in GGIR are the Euclidian Norm Minus One (ENMO) and Mean Amplitude Deviation (MAD) as these analytic techniques are perceived as not too complex for users and they have the ability of quantifying output in universal units instead of abstract scales [see S1 Table for more details] [8, 9, 12]. As the shift from using activity counts cut-points to classify activity intensities into raw accelerometer data processing is still evolving, a new metric that replicates the closed-source Actilife software was developed in the GGIR package, i.e. the "counts per minute" (CPM) metric [13]. This CPM metric has the ability to process data of the vertical axis (VA) only or to work with the vector magnitude (VM) [See S1 Table] [13]. The main advantage of this new metric in GGIR is the reduction of human errors. Data processing in ActiLife software requires manual processing of data to define wear and nonwear times, where the GGIR package applies the same nonwear algorithm on each data file [13]. Despite the popularity of working with accelerometer data, no gold standard exist for the most appropriate activity intensity-based cut-point accompanied by a data reduction metric. This lack of standardization affects the time spent in movement behavior and hampers comparability between studies [14].

Previous studies have highlighted that there are differences in cut-point dependent PA and SB durations depending on whether they are derived from raw accelerometer data or "counts per minute" data [14, 15]. In contrast, literature shows comparable findings for cut-point independent average acceleration and intensity gradient across the acceleration metrics ENMO and MAD [16]. Interestingly, although 24h-MBs are codependent, none of these studies interpreting time spent engaged in behaviors used a compositional behavioral approach but focused on one or more behaviors in isolation (e.g. PA, SB, sleep). Moreover, no previous studies have compared three different movement behavior features (i.e. time spent in a 24h period, overall activity volume, and overall activity intensity) between the new CPM metric for VA and VM, the ENMO metric and the MAD metric with hip-worn accelerometer data in adults.

Therefore, the objective of this study is threefold. First, we aimed to compare the movement behavior features resulting from commonly used accelerometer processing metrics among adults (i.e. ENMO, MAD, CPM VA and CPM VM). Second, we will investigate how these metrics affect the prevalence of meeting or not meeting the movement behavior guidelines for adults. Third, we aimed to explore whether the associations between movement behavior features and cardiometabolic variables differ according to the choice of metric. These aims can provide valuable insights into how different metrics can impact cut-point dependent and cut-point independent movement behaviors as well as how this affects predictions of health, which can in turn help in interpreting and comparing data analyzed in other studies.

## Methods

### Participants and procedure

This cross-sectional study used a Belgian sample of adults aged 25 to 64 years who were employed for at least 50% per week and had no physical (e.g. amputations, paralysis, recent stroke), cognitive (e.g. dementia, psychological disorders) or major medical (e.g. chronic respiratory diseases, heart failure) conditions that obstruct daily functioning. This study was approved by the ethical committee of Ghent University Hospital, and all participants provided written informed consent prior to the study (ONZ-2022_0013). Participants visited Ghent University Hospital once between 18[th] of April 2022 and 28[th] of March 2023. The following variables were measured during the study visit: 24h-MBs, cardiometabolic variables, and socio-demographic variables.

## Accelerometer-derived movement behavior features

To assess 24h-MBs, a tri-axial Actigraph wGT3x+BT was used to objectively quantify accelerations in orthogonal directions of a three-dimensional space [6]. Participants wore the accelerometer during the day on their right hip and at night on their nondominant wrist [6]. The durations of waking up and going to bed were recorded in a diary. When participants removed the device for water-based activities (e.g. swimming) or for other reasons (e.g. contact sports), the duration was recorded in the diary. The accelerometer was initialized via Actilife software and was set up to measure at a frequency of 100 Hz with 60-second epochs [6].

## Cardiometabolic variables

Fasting blood samples were collected to analyze glucose (mg/dL), HbA1c (mmol/mol), total cholesterol (mg/dL), high-density lipoprotein (HDL) cholesterol (mg/dL), low-density lipoprotein (LDL) cholesterol (mg/dL), and triglyceride (mg/dL) levels. The LDL-C (mg/dL) concentration was calculated as follows: LDL-C = total cholesterol–HDL-C–(triglycerides/5). Participants were instructed to refrain from eating eight hours before the visit. Blood pressure (BP), i.e. systolic BP (SBP) and diastolic BP (DBP), were measured twice (at an interval of one minute) in mmHg via an oscillometric device (OMRON M6 Comfort) on the right arm after 10 minutes of rest while the participant was in a seated position. Additionally, a TANITA SC-240MA scale was used to measure weight in kilograms (to the nearest 0.1 kg), body mass index (BMI;) in kg/m$^2$), and fat percentage (%). Height in meters (to the nearest 0.01 m) was measured by a Seca 213. Hip circumference and waist circumference (WC) were measured to the nearest 0.1 cm. Both measurements were used to calculate the waist-to-hip ratio, i.e. WHR = WC in cm/hip circumference in cm. All these variables were measured twice with the participants barefoot while wearing light clothes. A mean score was calculated for each variable. Finally, medication intake (names and class) was assessed using the Anatomical Therapeutic Chemical classification codes to classify medication as glucose-lowering medication, lipid-lowering medication or BP-lowering medication.

## Sociodemographic variables

Sociodemographic variables, including age, sex, educational level, smoking status, and pathology, were collected via a self-report questionnaire. Educational level was classified as low (primary or secondary school degree), middle (college degree) or high (university degree). Smoking status was classified as smoker, non-smoker or ex-smoker. Pathology was defined as having a diagnosis of a chronic condition (i.e. type 2 diabetes mellitus).

## Movement behavior feature analysis

Movement behavior features (i.e. mean time spent in the 24h-MBs, overall activity volume and intensity) were derived from raw accelerometer signals using the open source R package GGIR [9]. Accelerometer data were processed four times, i.e. for each metric separately, with a consistent GGIR script: 1) the ENMO metric, 2) the MAD metric, 3) the CPM VA metric, and 4) the CPM VM metric [See S1 Table] [12, 13]. The GGIR package uses an autocalibration algorithm that checks and corrects for calibration errors in triaxial accelerometer signals [12]. Actigraph files (n = 1) with a postcalibration error greater than 0.01 *g* were excluded [12]. Nonwear time was defined as a period of 60 minutes during which less than 13 m*g* for at least two out of three axes was noted or the range of accelerations accumulated to less than 50 m*g* [9]. Additionally, all the time periods indicated in the diary where participants removed the device were classified as nonwear time [6]. Accelerometer data were considered valid if the

device provided data for at least four days (including a minimum of three weekdays and one weekend day) with a minimum of 16 valid wear-time hours a day [17].

First, the ENMO metric (default metric in GGIR) was used to analyze the raw accelerometer data. The ENMO is calculated from the resultant vector of the measured orthogonal acceleration (three raw acceleration signals), adjusted for gravity by subtracting one gravitational unit and rounding to zero [8, 12, 18]. The cut-points of Hildebrand and colleagues [18] were used, i.e. light PA (LPA) (47 m*g*), moderate PA (MPA) (69 m*g*) and vigorous PA (VPA) (260 m*g*). Second, as the ENMO metric often suffers from calibration errors being too sensitive, the MAD metric (which works with average subtractions) seems to better account for offsetting signal noise [12]. MAD describes the distance of data points around the mean [19, 20]. The cut-points of Vaha Ypya and colleagues [19, 20] were used, i.e. LPA (22.5 m*g*), MPA (94 m*g*), and VPA (396 m*g*). Third, the new "activity counts" metric, which replicates the Actilife process based on the recently published paper, was used [13]. This metric works with CPM cut-points for the VA, as in Troiano and colleagues [21], i.e. LPA (100 CPM), MPA (2020 CPM), and VPA (5999 CPM), as well as for VM, by applying the cut-points of Sasaki and colleagues [22, 23], i.e. LPA (200 CPM), MPA (2690 CPM) and VPA (6166 CPM). These cut-points are epoch length specific and need to be corrected by a conversion factor before using within the GGIR package. As recommended, cut-points were divided by the epoch length in the new study divided by the epoch length in the original validation study, i.e., CPM*(5/60) [9]. All behaviors were represented in minutes a day (min/day) and weighted to represent an average day (i.e. ((weekdays*5)+weekend days*2))/7).

In addition to the mean time spent in different intensities of movement behaviors, the overall activity volume and intensity gradient were calculated [10]. The overall activity volume is the average acceleration accumulated in a 24h day represented in m*g*. The intensity gradient and intercept refer to the intensity distributed over a 24h day. This is represented by an intercept and gradient (slope) of the linear regression between the log of daily intensity and the log of time in that intensity. A smaller intensity gradient (more negative, steeper slope) reflects a more sedentary profile [10]. Both are directly measured and independent from population-specific intensity-based cut-point validation studies. The average acceleration and intensity gradient are moderately correlated with each other, providing insights into the amount of activity or the intensity of the activity performed during a 24h day [10].

Sleep was calculated by the GGIR package using the time needed to wake up and go to bed, as reported in the sleep diary. In the case of invalid sleep diary data, the HDCZA algorithm was used to detect sleep period. The HDCZA algorithm detects the sleep period by searching for periods of time during which the z-angle does not change by more than 5 degrees for at least 5 minutes [24].

## Statistical analysis

Participant characteristics are presented as the means and standard deviations (± SDs) for continuous data and proportions (%) for categorical data.

First, the single movement analysis included a intraclass correlation coefficient (ICC) estimates, including 95% confidence intervals (single rater, absolute agreement, two way random) and Bland–Altman plots (mean difference (±SD), limits of agreements, mean absolute percentage error), to determine differences between the metrics for 1) time spent in individual movement behaviors (single movement analysis), 2) average acceleration and 3) intensity gradient [25]. Due to the use of the same processing technique for sleep (each metric applies the same sleep algorithm) making this comparative analysis for sleep redundant. Additional distribution plots of these movement behavior features are presented in S1 Fig.

Compositional Data Analysis (CoDA) was used to account for the codependency of the 24h-MBs using the R packages *compositions* and *codaredistlm* [26, 27]. The 24h-MBs compositions created by each accelerometer metric (consisting of sleep, SB, LPA, and MVPA) were expressed as four sets of three isometric log-ratios (ILRs) [28]. Variation matrices were created to explore the variance covariance of the 24h-MBs [see S2 Table] [28]. Linear mixed effects models were used to explore differences between compositions. The dependent variables were the ILRs, in long 'stacked' format with a dummy variable indicating whether they were ILR1, ILR2 or ILR3. As described in Lim and colleagues [29], random slopes were added at the log ratio level, grouped by participant ID, and repeated within participant ID (random intercept) for each accelerometer metric composition. Fixed effect interactions between the metrics and the dummy variable representing the ILR number were analyzed to test for differences between the different accelerometer metrics (MANOVA F test). Full models included additional interactions to adjust for covariates.

Second, compliance with the 24h-MBs guidelines was calculated for each metric. Adults were classified into one of the following categories: compliance with (1) no guidelines, (2) PA guideline, (3) SB guideline, (4) sleep guideline, (5) PA+SB guidelines, (6) PA+sleep guidelines, (7) SB+sleep guidelines, and (8) all three guidelines. Compliance with the guidelines was defined as a sleep duration between 7 and 9 hours a day, sedentary time limited to 8 hours a day and/or MVPA for 30 minutes a day [2, 30].

Third, linear regression models were fitted for each metric separately to explore associations between the 24h-MBs composition, average acceleration and intensity gradient as independent variables and cardiometabolic variables as the dependent variable. The in-depth analysis of these models are reported in the S1 File. Model assumptions of linearity, normality of residuals, posterior predictive checks, influential observations, collinearity and homogeneity of variance were evaluated using the *performance* package [31]. If the linear model did not meet the assumptions, log-transformed cardiometabolic variables were used in the analysis. The estimates, t-value, p-value and adjusted $R^2$ were reported for associations between the average acceleration and intensity gradient on the one hand and cardiovascular variables on the other hand.

For the 24h-MBs composition the interpretation of the strength and directions of associations are plotted as time reallocations models. These predictions estimated the average difference in cardiometabolic health outcomes when time (e.g. -20 to +20 minutes) in one behavior was proportionally exchanged with time in the remaining behaviors. To enhance the interpretability of the time reallocation models, the log-transformed data were back-transformed to their raw units prior to computing differences. The outcomes of the time reallocations are presented as the absolute differences between the estimated and the mean cardiometabolic variable and the standardized effect size (ES), which is the absolute difference divided by the standard deviation of the particular variable.

All models were adjusted for sex, age, educational level, smoking status, medication intake, and pathology. Complete-case analysis was used in all models. All analyses were performed in R version 4.1.1 [32]. A p value <0.05 was considered to indicate statistical significance.

## Results

This study included data from 213 adults, 68.5% (n = 146) of whom were female, with a mean age of 45.8 (SD = 10.8) years. Eighty-two adults (38.5%) were classified as normal weight (18–24.99 kg/m$^2$), 80 adults (37.6%) as overweight (25–29.99 kg/m$^2$) and 51 adults (23.9%) as obese (≥30 kg/m$^2$). Of the total sample of 213 adults, 22 adults were diagnosed with type 2 diabetes mellitus (10.3%). The mean wear time was 1424 min/day (SD = 38 min/day), and the

mean number of valid days was 5.8 days (SD = 0.4). See Table 1 for additional participant characteristics.

## Comparison of movement behavior features

The single movement analysis exploring the time use of each behavior separately showed poor agreement between the ENMO and any other metric for SB and LPA (ICC < 0.5, p<0.001). Poor agreement was found between the CPM VM and MAD for LPA, as well as between the CPM VA and MAD for MVPA (ICC < 0.5, p<0.001). Moderate to good agreement was found between the CPM VA and the CPM VM for SB, LPA and MVPA (ICC 0.53–0.78, p<0.001). Additionally, moderate to good agreement was found between the CPM VA and MAD for SB and LPA (ICC 0.68–0.89, p<0.001) as well as between the CPM VM and MAD for SB and MVPA (ICC 0.55–0.77, p<0.001). Finally, moderate to good agreement was found between the ENMO and any other metric for MVPA (ICC 0.61–0.83, p<0.001). For the average acceleration and intensity gradient, poor agreement was found between all the metrics (ICC < 0.5, p<0.001) except for the CPM VA versus MAD (ICC 0.70–0.84, p<0.001). Bland–Altman plots revealed the greatest difference between ENMO and CPM VM, with wide limits of agreement for SB (bias of +334 minutes) and LPA (bias of -325 minutes). The smallest limits of agreement were found when comparing the CPM VA and MAD for SB and the CPM VA versus the CPM VM for LPA (bias of -25 minutes and -52 minutes, respectively). Regarding MVPA, the widest limits of agreement were found when comparing CPM VA with MAD (bias of -32 minutes), and the smallest limits were found for CPM VM versus MAD (bias of -6 minutes). The average acceleration and intensity gradient showed the widest limits of agreement for ENMO versus CPM VM (bias of -44.5 m*g* for average acceleration and -0.6 for intensity gradient) and the smallest for CPM VA versus MAD (bias of -2.4 m*g* for average acceleration and +0.1 for intensity gradient) [S3 Table and Fig 1].

Considering the codependency of behaviors, the compositional analysis showed different results according to the metrics. Significant differences were found between the four metrics for the mean time spent on 24h-MBs (all p<0.001) [S4 Table]. Using ENMO resulted in 24h-MBs compositions with the highest proportion of SB (59% for ENMO compared to 42% for MAD, 40% for CPM VA and 35% for CPM VM). Regarding LPA, the CPM VM had the highest percentage of time spent in the LPA (25% for the CPM VM compared to 2% for the ENMO, 17% for the MAD 22% for CPM VA). Finally, the highest proportion of MVPA in a 24h-MBs composition was found for MAD (5%), whereas lower proportions were found for ENMO (3%), CPM VA (2%) and CPM VM (4%) [see Table 1].

## Comparison of guideline compliance

All accelerometer measurements reported low compliance rates for the integrated guidelines (i.e. complying with three guidelines), ranging from 0 to 6%, except for the CPM VM metric, for which 25% of the adults complied with the three guidelines. With any of the guidelines, 15% of adults were classified as noncompliers according to the ENMO and CPM VA, whereas 2 to 4% of adults were classified as noncompliers according to the MAD and CPM VM [S5 Table].

## Comparison of associations between movement behavior features and cardiometabolic variables

The time spent on 24h-MBs was significantly associated with BMI and HbA1c when ENMO was used (BMI: F = 3.23, p = 0.02; HbA1c: F = 2.80, p = 0.04). Additionally, 24h-MBs compositions were significantly associated with WC when using ENMO or MAD (F = 3.11, p = 0.03;

**Table 1. Sociodemographic, cardiometabolic and movement behavior characteristics of the total sample.**

| | Total sample (n = 213) |
|---|---|
| Sex = female (n (%)) | 146 (68.5) |
| Age (mean (SD)) | 45.8 (10.8) |
| Education (n (%)) | |
| Low | 51 (24.3) |
| Middle | 90 (42.9) |
| High | 69 (32.9) |
| *Missing (n (%))* | *3 (1.4)* |
| Smoking (n (%)) | |
| Ex-smoker | 33 (15.6) |
| non-smoker | 164 (77.7) |
| Current smoker | 14 (6.6) |
| *Missing (n (%))* | *2 (0.9)* |
| BMI—kg/m$^2$ (mean (SD)) | 27.3 (5.5) |
| BMI category (n (%)) | |
| Normal weight | 82 (38.5%) |
| Overweight | 80 (37.6%) |
| Obesity | 51 (23.9%) |
| WHR (mean (SD)) | 0.9 (0.1) |
| WC—cm (mean (SD)) | 94.8 (14.9) |
| Fat % (mean (SD)) | 31.7 (9.2) |
| *Missing (n (%))* | *22 (10.3)* |
| SBP—mmHG (mean (SD)) | 122.1 (15.2) |
| DBP—mmHG (mean (SD)) | 79.8 (9.5) |
| Glucose lowering medication = yes (n (%)) | 22 (10.3) |
| Lipid lowering medication = yes (n (%)) | 32 (15.0) |
| Blood pressure lowering medication = yes (n (%)) | 38 (17.8) |
| HbA1c - mmol/mol (mean (SD)) | 36.6 (6.3) |
| *Missing (n (%))* | *12 (5.6)* |
| Glucose—mg/dL (mean (SD)) | 89.2 (19.3) |
| *Missing (n (%))* | *6 (2.8)* |
| Total cholesterol—mg/dL (mean (SD)) | 189 (34.7) |
| *Missing (n (%))* | *3 (1.4)* |
| HDL-Cholesterol—mg/dL (mean (SD)) | 57.6 (12.8) |
| *Missing (n (%))* | *3 (1.4)* |
| LDL-Cholesterol—mg/dL (mean (SD)) | 111.7 (30.5) |
| *Missing (n (%))* | *3 (1.4)* |
| Triglycerides—mg/dL (mean (SD)) | 101.8 (78.1) |
| *Missing (n (%))* | *3 (1.4)* |
| ENMO metric | |
| Sleep* (min/day—%) | 512 (36%) |
| SB* (min/day—%) | 854 (59%) |
| LPA* (min/day—%) | 34 (2%) |
| MVPA* (min/day—%) | 40 (3%) |
| Average acceleration (m*g*) (mean (SD)) | 18.66 (5.74) |
| Intensity gradient (mean (SD)) | -2.22 (0.28) |
| Intensity intercept (mean (SD)) | 12.34 (0.95) |
| Intensity R$^2$ (mean (SD)) | 0.89 (0.07) |

*(Continued)*

**Table 1.** (Continued)

|  | Total sample (n = 213) |
|---|---|
| MAD metric |  |
| Sleep* (min/day—%) | 516 (36%) |
| SB* (min/day—%) | 608 (42%) |
| LPA* (min/day—%) | 250 (17%) |
| MVPA* (min/day—%) | 66 (5%) |
| Average acceleration (m*g*) (mean (SD)) | 31.15 (9.29) |
| Intensity gradient (mean (SD)) | -1.89 (0.22) |
| Intensity intercept (mean (SD)) | 11.50 (0.84) |
| Intensity $R^2$ (mean (SD)) | 0.87 (0.07) |
| CPM VA metric |  |
| Sleep* (min/day—%) | 516 (36%) |
| SB* (min/day—%) | 581 (40%) |
| LPA* (min/day—%) | 309 (22%) |
| MVPA* (min/day—%) | 34 (2%) |
| Average acceleration (m*g*) (mean (SD)) | 28.96 (10.43) |
| Intensity gradient (mean (SD)) | -1.76 (0.19) |
| Intensity intercept (mean (SD)) | 10.89 (0.80) |
| Intensity $R^2$ (mean (SD)) | 0.88 (0.07) |
| CPM VM metric |  |
| Sleep* (min/day—%) | 517 (36%) |
| SB* (min/day—%) | 498 (35%) |
| LPA* (min/day—%) | 364 (25%) |
| MVPA* (min/day—%) | 61 (4%) |
| Average acceleration (m*g*) (mean (SD)) | 63.58 (18.12) |
| Intensity gradient (mean (SD)) | -1.56 (0.12) |
| Intensity intercept (mean (SD)) | 10.56 (0.62) |
| Intensity $R^2$ (mean (SD)) | 0.83 (0.06) |

* Time spent in movement behaviors is part of a composition where variation matrices are plotted as traditional variance–covariance [S2 Table]. SB: time spent in sedentary behavior, LPA: time spent in light physical activity, MVPA: time spent in moderate to vigorous physical activity, m*g*: milligravitational unit, Average acceleration: proxy of total volume of PA. The intensity gradient and intercept refer to the intensity distributed over a 24h day. This is represented by the intercept and gradient (slope) of the linear regression between the log of daily intensity and the log of time in that intensity. Intensity $R^2$: $R^2$ of the intensity gradient regression line. BMI: body mass index, WHR: waist-to-hip ratio, WC: waist circumference, SBP: systolic blood pressure, DBP: diastolic blood pressure, HDL: high-density lipid, LDL: low-density lipid. The Intensity-based cut-points thresholds for each metric are as follows: ENMO Hildebrand et al. (2014), MAD Vaha Ypya et al. (2018, 2023), CPM VA Troiano et al. (2008), and CPM VM Sasaki et al. (2011).

F = 2.75, p = 0.04, respectively). All 24h-MBs compositions, except for CPM VA, were significantly associated with fat% (ENMO F = 3.20, p = 0.02; MAD F = 2.68, p = 0.05; CPM VM F = 3.27, p = 0.02). For average acceleration, only MAD had a significant negative association with BMI (t = -2.52, p = 0.01) and fat% (t = -2.03, p = 0.04), and a positive association with WC (t = 3.68, p<0.001). The intensity gradient was negatively associated with BMI for ENMO (t = -1.98, p = 0.05), MAD (t = -2.72, p<0.01) and CMP VA (t = -2.48, p = 0.01), as well as with WC for MAD (t = -2.27, p = 0.02). However, the intensity gradient was positively associated with BMI (t = 2.53, p = 0.01) and WC (t = 2.5, p = 0.01) for CPM VM. See S1 File for more in-depth analysis of these linear models.

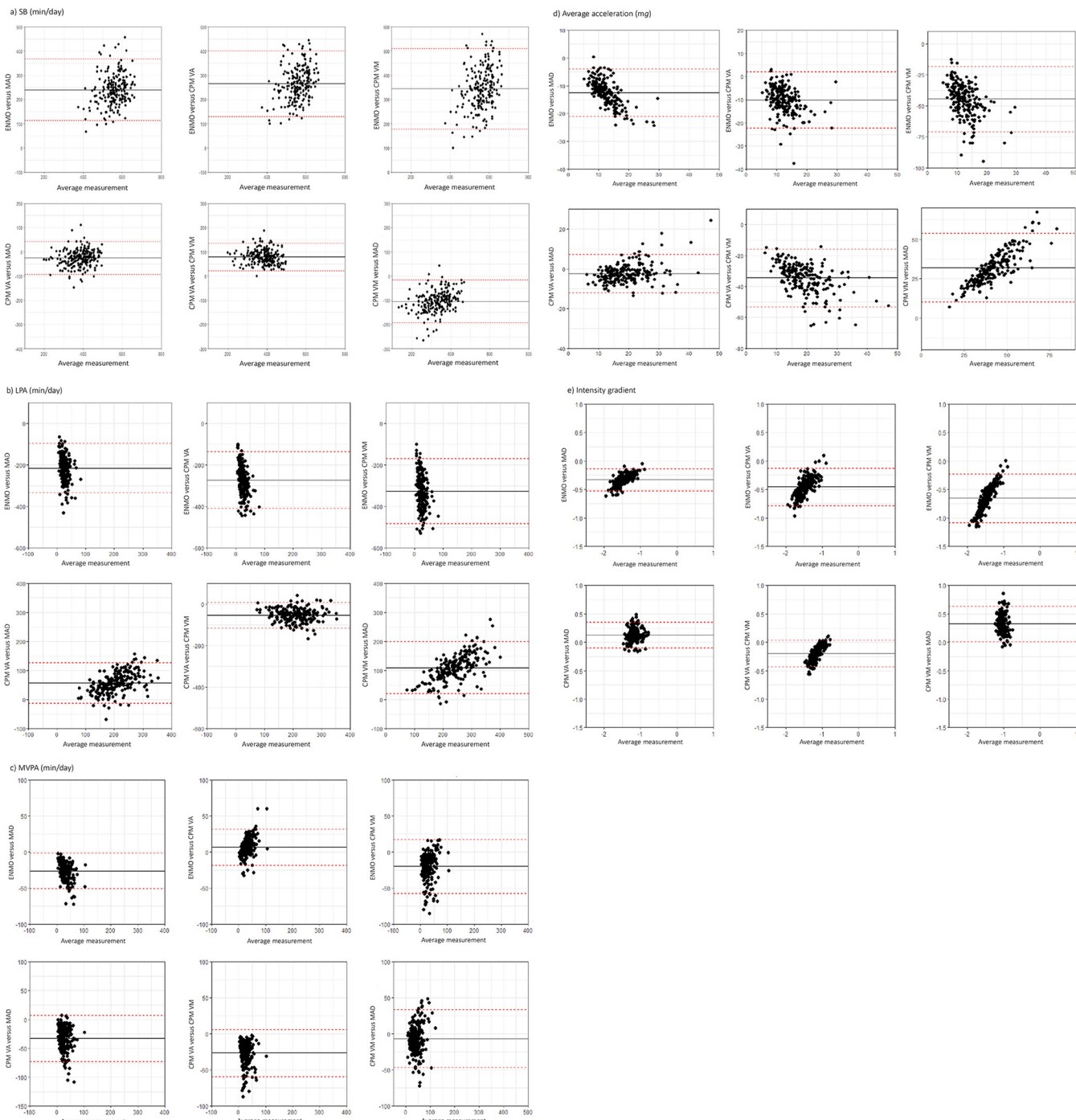

**Fig 1. Bland–Altman plots presenting the time use of single movement behaviors (SB, LPA, MVPA), average acceleration and intensity gradient according to four metrics (ENMO, MAD, CPM VA, CPM VM).** a) SB (min/day), b) LPA (min/day), c) MVPA (min/day), d) Average acceleration (m*g*), d) Intensity gradient. The average measurements presented on the x-axis refer to the average of metric A and metric B, i.e. from left to right, ENMO versus MAD, ENMO versus CPM VA, ENMO versus CPM VM, CPM VA versus MAD, CPM VA versus CPM VM, and CPM VM versus MAD. Black line: average difference/bias between metrics; red lines: upper and lower limits of agreement. The Intensity-based cut-points thresholds for each metric are as follows: ENMO Hildebrand et al. (2014), MAD Vaha Ypya et al. (2018, 2023), CPM VA Troiano et al. (2008), and CPM VM Sasaki et al. (2011).

## Discussion

This study revealed that the duration of daily movement behaviors, prevalence of guideline compliance and relationship between movement behaviors and cardiometabolic variables differed depending on the metric used to analyze accelerometry data.

Differences between metrics were found in determining movement behavior features, including cut-point-related times spent in single movement behaviors and cut-point-independent features such as average acceleration and intensity gradient. The ENMO metric produced the most sedentary time-use profile, average acceleration and intensity gradient, while the CPM VM metric produced the most active time-use profile. CPM VA and MAD demonstrated moderate to good agreement for all the features (time spent in SB and LPA, average acceleration, intensity gradient), except for time spent in MVPA. When considering the codependency of behaviors, all the 24h-MBs compositions were significantly different depending on the metric used. Thus, the choice of reduction metric to process accelerometry data can lead to different results and conclusions. As indicated by previous research in children and adults, differences found in time-use estimates are attributable to intensity-based cut-points accompanied by the metric [14, 15, 33–35]. Our study showed that cut-point independent features (i.e. average acceleration and intensity gradient) can also exhibit poor agreement between most metrics. Despite the main advantage of comparability across cohorts and accelerometers, caution is warranted when comparing them across metrics (ENMO, MAD, CPM VA, CPM VM) [10]. As our study did not attempt to validate one method above another, we are unable to make recommendations about metric selection. Nevertheless, in our study, the ENMO metric (Hildebrand and colleagues cut-point [18]) led the most disparate, and perhaps unrealistic, estimates of SB and LPA. Future research is necessary to further explore the most appropriate metric and cut-point for hip-worn data.

The discrepancies in estimates of time spent in movement behaviors, as measured by various metrics, have direct implications for estimates of compliance with the 24h-MBs guidelines. This observation aligns with existing research emphasizing the influence of the chosen cut-point on the prevalence of compliance [14, 34]. Previous research has reported a 7% full compliance rate among adults using CPM cut-points, where people who complied with all three guidelines had more favorable health parameters, such as BMI, WC, triglycerides, insulin, and glucose levels [1, 36]. Movement behavior guidelines are considered comprehensible for the general population, but in research, we must be aware of the rigidity of such guidelines and the potential consequences of categorizing individuals based on minute deviations. Cut-point dependent time-use estimates are features that are easily interpretable in compliance with the established guidelines, however, the cut-point independent average acceleration and intensity gradients are not. Therefore, recent research has attempted to improve the interpretability and ease of use of these features [37]. Rowlands et al. (2021) proposed a preliminary minimal clinically detectable difference recommendation of 1 m$g$ (comparable with 5 minutes of brisk walking) in daily average acceleration for wrist-worn data to gain health benefits among inactive adults [38]. Nevertheless, additional research is needed to confirm these results [38]. Moreover, Schwendinger et al. (2023) developed reference values and percentile curves for wrist-worn accelerometer data to use average acceleration and intensity gradient estimates in healthy adults [11]. Since accelerations are known to differ depending on the wear location (e.g. hip versus wrist) [11, 15, 39], the results of this study could not be compared with these percentile curves. Future research should look into developing similar percentile curves for hip-worn accelerometry. Despite the absence of reference values for hip-worn accelerometer data, the strength of the correlation between average acceleration and intensity, as well as the duration of LPA or MVPA, provides additional insights into health associations related to both the

quantity and intensity of activity [10, 37]. The independence of the intensity gradient compared to cut-point dependent time use at different intensities has previously been reported, evidenced by a smaller magnitude of correlation between the intensity gradient and LPA and MVPA compared to a higher correlation between intensity gradient and average acceleration [10, 40]. Moreover, both the intensity gradient and average acceleration were more strongly associated with MVPA than with LPA, indicating a better capture of higher intensities [10, 40].

While the general trends in associations between 24h-MBs composition and cardiometabolic health were similar across all metrics, the strength of associations and type of behavior involved in the associations varied depending on the metric used. The ENMO metric showed the most associations with different cardiometabolic health variables (BMI, WC, HbA1c and fat%), whereas MAD was only associated with WC and fat% and CPM VM was only associated with fat%. Other research has shown significant associations for BMI, WC and fat% when MAD and CPM VA were used [41–43]. Additionally, the behavior within the total composition (sleep, SB, LPA, MVPA) that was significantly associated with a cardiometabolic health variable differed depending on the metric used. In general, MVPA predicted the strongest health improvements, and these predictions were most consistent across metrics, except for some inconsistent results for fat%. These results are comparable with a recent review highlighting the greatest health effects when reallocating time toward MVPA, where evidence for reallocating time into LPA as well as out or into sleep is more inconclusive [44]. In this study, the composition retrieved from the ENMO metric seemed to have unrealistic results regarding time spent in LPA and SB which in turn might affect the associations. Furthermore, differences between metrics were found for associations with cut-point independent features (average acceleration and intensity). All the metrics showed significant associations with the intensity gradient and BMI, but the directions of association were different. As studies reporting the association between these newer cut-point independent movement behavior features and cardiometabolic health are limited [45, 46], this paper emphasizes the potential impact of the selected metrics when comparing results with other research.

This is the first study comparing average acceleration and intensity gradients derived from different metrics among an adult population (n = 213). Additionally, this is the first study comparing the new CPM metric developed by GGIR developers to replicate the Actilife process [13]. Perfect reproducibility across the four metrics was ensured by the use of the GGIR package, which allows for consistent data reduction features, i.e. autocalibration, sleep algorithm, and nonwear detection methods. For each metric a commonly used cut-point was chosen to classify activity intensities. Although commonly used cut-points were selected, other cut-points are available, which are all based on a specific validation protocol in a specific sample. Using other cut-points might provide other results. Future research should focus on newer Machine Learning Techniques to classify movement behavior patterns, which can align accelerometer data with for example heart rate variability to accurately classify activities [47]. Next, adults were categorized as noncompliant with SB guidelines if they exceeded the threshold of 8 hours per day, as per the Canadian Society of Exercise Physiology guidelines. However, the use of the specific threshold has been criticised as it is largely underpinned by cross-sectional evidence [2]. Therefore guideline compliance in this paper should be interpreted with caution. Finally, only hip-worn data for the waking day were used in this study, which hinders comparison with studies using wrist-worn data due to differences in acceleration based on body location [15].

## Conclusion

Depending on the chosen metric, differences were found for cut-points dependent (time use estimates) and cut-point independent (average acceleration and intensity gradient) movement

behavior features. The ENMO metric classified adults with the most sedentary behavior profiles as CPM VM metric had the highest activity profiles, both with implications for guideline compliance prevalence and associations with cardiometabolic variables. However, classification of SB and LPA seemed the least realistic when the ENMO metric was used. Researchers should be aware of the implications of metric choice in data processing for data interpretation and comparability across studies.

## Supporting information

**S1 Fig. Distribution plots of movement behavior features.** SB: sedentary behavior, LPA: light physical activity, MVPA: moderate to vigorous physical activity, ENMO: Euclidian Norm Minus One, MAD: Mean Amplitude Deviation, CPM VA: Counts Per minute Vertical Axis, CPM VM: Counts Per Minute Vector Magnitude.
(PDF)

**S1 Table. Table with accelerometer data processing metrics used to analyze movement behavior features.**
(DOCX)

**S2 Table. Variation matrix representing the codependency between the 24h-MBs among adults.** SB: sedentary behavior, LPA: light physical activity, MVPA: moderate to vigorous physical activity; a variation value close to zero indicates that two behaviors are highly proportional, i.e. codependent, which means that as one behavior varies, the other behavior similarly increases or decreases. The intensity-based cut-points thresholds for each metric are as follows: ENMO Hildebrand et al. (2014), MAD Vaha Ypya et al. (2018, 2023), CPM VA Troiano et al. (2008), and CPM VM Sasaki et al. (2011).
(XLSX)

**S3 Table. Intraclass correlations and mean differences between different metrics regarding single movement time use, average acceleration and intensity gradient.** ICC (95% CI): intraclass correlation coefficient (95% confidence interval), MD: mean difference between two metrics, SD: standard deviation of the mean difference, LOA: limit of agreement, MAPE: mean absolute percentage error. The intensity-based cut-points thresholds for each metric are as follows: ENMO Hildebrand et al. (2014), MAD Vaha Ypya et al. (2018, 2023), CPM VA Troiano et al. (2008), and CPM VM Sasaki et al. (2011).
(XLSX)

**S4 Table. Significant differences in 24h-MBs composition according to four metrics (ENMO, MAD, CPM VA, and CPM VM).** Linear mixed models were used to take into account the compositional data represented by isometric log ratios per composition stacked within the participant The intensity-based cut-points thresholds for each metric are as follows: ENMO Hildebrand et al. (2014), MAD Vaha Ypya et al. (2018, 2023), CPM VA Troiano et al. (2008), and CPM VM Sasaki et al. (2011).
(XLSX)

**S5 Table. Compliance with 24h-MBs guidelines according to four metrics (ENMO, MAD, CPM VA, CPM VM).** SB: sedentary behavior, LPA: light physical activity, MVPA: moderate to vigorous physical activity. The Intensity-based cut-points thresholds for each metric are as follows: ENMO Hildebrand et al. (2014), MAD Vaha Ypya et al. (2018, 2023), CPM VA Troiano et al. (2008), and CPM VM Sasaki et al. (2011).
(XLSX)

**S1 File. Associations between time use features and cardiometabolic variables.** This document contains a more in-depth analysis to explore the impact of the metric on the associations between the cut-point dependent movement behaviors features (time use estimates in SB, LPA, MVPA) and the cut-point independent movement behaviors (average acceleration, intensity gradient) on one hand and the cardiometabolic variables on the other hand. (DOCX)

**S2 File. Configuration file for data processing in GGIR regarding ENMO.** The same configuration was replicated for MAD, Counts Per Minute Vetrical Axis, Counts Per Minute Vector Magnitude. See S1 Table for different specifications in GGIR regarding each metric. (CSV)

## Acknowledgments

We would like to thank all participants included in this study.

## Author Contributions

**Conceptualization:** Iris Willems.

**Data curation:** Iris Willems.

**Formal analysis:** Iris Willems.

**Investigation:** Iris Willems.

**Methodology:** Iris Willems.

**Supervision:** Marieke De Craemer.

**Visualization:** Iris Willems.

**Writing – original draft:** Iris Willems.

**Writing – review & editing:** Iris Willems, Vera Verbestel, Dorothea Dumuid, Patrick Calders, Bruno Lapauw, Marieke De Craemer.

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
