## [Decision Letter · Decision Letter 0]

19 Jun 2024

PONE-D-24-08044A comparative analysis of 24-hour movement behaviors features using different accelerometer metrics in adults: implications for guideline compliance and associations with cardiometabolic healthPLOS ONE

Dear Dr. De Craemer,

Thank you for submitting your manuscript to PLOS ONE. After careful consideration, we feel that it has merit but does not fully meet PLOS ONE’s publication criteria as it currently stands. Therefore, we invite you to submit a revised version of the manuscript that addresses the points raised during the review process.

**ACADEMIC EDITOR: ****Dear Author, Please make the necessary changes as suggested by the reviewers to make you manuscript more sound scientifically.** The decision of this manuscript is justified based on PLOS ONE’s publication criteria and not on its novelty or perceived impact.==============================

We look forward to receiving your revised manuscript.

Kind regards,

Zulkarnain Jaafar

Academic Editor

PLOS ONE

Journal Requirements:

Reviewers' comments:

Reviewer's Responses to Questions

**Comments to the Author**

1. Is the manuscript technically sound, and do the data support the conclusions?

Reviewer #1: Yes

Reviewer #2: Yes

2. Has the statistical analysis been performed appropriately and rigorously? 

Reviewer #1: Yes

Reviewer #2: I Don't Know

3. Have the authors made all data underlying the findings in their manuscript fully available?

Reviewer #1: Yes

Reviewer #2: Yes

4. Is the manuscript presented in an intelligible fashion and written in standard English?

Reviewer #1: Yes

Reviewer #2: Yes

5. Review Comments to the Author

Reviewer #1: The results indicate that the movement behavior data varied depending on the metric used for analysis. Specifically, ENMO (Euclidean Norm Minus One) represented the most sedentary movement behavior profile, while CPM (Counts Per Minute) vector magnitude represented the most active profile.

This suggests that different accelerometer metrics capture different aspects of movement behavior, with some metrics highlighting more sedentary patterns while others emphasize more active behaviors. Understanding these differences is crucial for accurately assessing individuals' activity levels and sedentary behavior, which in turn can inform interventions aimed at promoting physical activity and reducing sedentary time to improve overall health outcomes.

Interestingly, the study found that reallocating time towards moderate-to-vigorous physical activity consistently predicted significant improvements in cardiometabolic variables, with the exception of fat percentage. This suggests that increasing time spent in moderate-to-vigorous physical activity may have positive effects on various aspects of cardiometabolic health, highlighting the importance of engaging in activities that elevate heart rate and promote greater exertion.

Overall, these findings emphasize the complexity of studying movement behaviors and their associations with health outcomes, indicating that the choice of accelerometer metric can influence both compliance rates with guidelines and the observed relationships with cardiometabolic variables.

Overall, the study suggests variations in agreement between different metrics, with some showing better consistency than others across various activity intensities. These findings underscore the importance of considering the choice of metric carefully when analyzing accelerometer data for physical activity assessment.Average acceleration showed strong associations with MVPA across all metrics, its relationship with LPA was weaker. Additionally, the intensity gradient exhibited stronger correlations with MVPA compared to LPA, with some variation across metrics.

The choice of metric significantly influences the composition of 24h MBs, particularly in terms of the distribution of time spent in sedentary behavior, light physical activity, and moderate to vigorous physical activity. ENMO tended to allocate more time to sedentary behavior, while CPM VM favored light physical activity, and MAD had a higher proportion of moderate to vigorous physical activity.

The analysis revealed significant associations between time spent on 24h movement behaviors (MB) and various cardiometabolic variables, with differences observed across different accelerometer metrics. Overall, these findings underscore the importance of considering different accelerometer metrics when examining associations between physical activity behaviors and cardiometabolic health outcomes. The direction and magnitude of associations varied across metrics, highlighting the need for tailored interventions aimed at promoting specific types of physical activity to improve cardiometabolic health.

The study has several limitations that should be considered when interpreting the results:

Selection of Cutoff Points: The use of cutoff points for each accelerometer metric is a potential limitation. While commonly used cutoff points were selected, there are alternative cutoff points available. These cutoff points are typically based on specific validation protocols conducted in particular populations. Therefore, the choice of cutoff points may impact the interpretation of physical activity data and comparisons across studies.

Limited Generalizability: The study only used hip-worn accelerometer data for the waking day. This may limit the generalizability of the findings, particularly when comparing them to studies using wrist-worn accelerometer data. Differences in acceleration patterns based on body location can influence the assessment of physical activity levels. Therefore, caution is needed when generalizing findings to populations or studies using different accelerometer placements.

Sample Characteristics: The results may be influenced by the characteristics of the sample population studied. Demographic factors such as age, gender, and physical fitness levels can affect physical activity patterns and associations with health outcomes. Therefore, the findings may not be representative of other populations with different demographic profiles.

Cross-Sectional Design: The study likely employed a cross-sectional design, which limits the ability to establish causal relationships between physical activity behaviors and cardiometabolic health outcomes. Longitudinal studies are needed to better understand the temporal relationships between these variables and to assess the effectiveness of interventions.

Measurement Error: Accelerometer measurements are subject to measurement error, which can arise from factors such as device malfunction, wear time compliance, and data processing methods. These errors could potentially affect the accuracy and reliability of the physical activity measurements and subsequent associations with health outcomes.

Acknowledging these limitations can help researchers and clinicians better interpret the study findings and guide future research efforts aimed at addressing these limitations to improve the understanding of physical activity and its impact on health.

Reviewer #2: This paper focuses on the different metrics available in the GGIR package for accelerometer processing. This is quite a technical aspect of physical activity measures, but an important one, especially in light of the findings which show that the different metrics can produce quite different summary estimates. In general, I think this is a good, well-written, useful paper. However there are areas which are unclear and I feel it currently tries to tackle too much. The discussion would also benefit from less focus on just repeating the results and more on what this means for someone about to begin a study – which metric should they use and why? I list some more detailed points below.

Major points

1) Currently, definitions and advantages and disadvantages of the different processing metrics and PA summary measures don’t appear until the methods section (eg ENMO at line 142, intensity gradient line 160+), but this makes it very hard to understand the background and why these different metrics and summaries (and hence the manuscript itself) are important. As this manuscript involves very technical aspects of accelerometer processing which not all readers will be familiar with, I suggest adding a clear overview of the full process (raw data to processed data to cut points & summaries) early on in the background. Then describe the different processing metrics and the different PA measures - definitions, where they fit into this process, how they differ and how it might be expected to affect the summaries.

2) This manuscript is trying to do a lot of things, which makes it difficult to follow. In particular, associations between PA measures and cardiovascular outcomes seem out of place – they’re not affected directly by the processing metrics, the analysis is not in-depth enough to be a full association study and its not clear what the implications are for different metrics. I think the paper would be stronger and clearer if this section were dropped entirely. If the authors do decide to keep this, then there needs to be more linking to the metrics and crucially some guidance on the appropriate metrics to use in different circumstances.

3) Much of the model description in the methods is not clear, so I am unable to comment on the suitability of the modelling. For example, at line 192 – what’s the outcome in this model? What fixed effect terms are included? What are the random terms? Line 192 refers to a single model, but line 194 to multiple models –how many models and what are they? Also, line 193 refers to random slopes, but line 194 talks about random intercept models (ie random intercept only) – which is it? Finally, I think there might be repeated measures on the same individual in here – is there a random intercept for participant included? A model equation would be useful, eg in the Supporting Information.

Minor points

Abstract

• Line 5 : The abbreviations ENMO and MAD need definitions

• Line 9: in what country?

Background

• Lines 35-41: The ‘combined’ guidelines mentioned here are new and I’m not sure if they extend beyond Canada - most countries and the WHO only have guidelines for physical activity. I think it’s worth being clear that these are not internationally-agreed guidelines, and in fact the WHO 2020 guidance explicitly says that ‘evidence was insufficient to quantify a sedentary behaviour threshold’.

• Lines 61-62: I’m not sure what point is being made here. Regardless of metrics, if the accelerometer is removed, determining non-wear time will be a problem. If not, then determining sleep time is an issue.

Methods

• Line 89-90: what does ‘active on the job market’ mean? Is this employed, or unemployed and looking for work? (to me, the ‘job market’ refers to recruitment for new posts)

• Line 97: more details required. How many days? Including weekends? Was there a minimum wear time to be included (either per day, or number of days?)

• Line 124-5: can you give more detail of the educational system for the international reader eg ages, qualifications. I don’t understand the distinction between ‘higher education degree’ and ‘university degree’.

• Line 136-8: add reference or justification

• Lines 142+ this information on metrics belongs in the background section

• Line 149-155 : mixing of reference styles

• Line 178: add reference

• Line 183-185: It’s not clear, but either these are the same MB measures for different metrics, in which case they are measuring the same thing so no surprise if they’re correlated, or they’re the same metric and different MBs, in which case they must be correlated because they sum to 24h. Either way, please justify why correlations are useful/relevant.

• Lines 214-7 for log-transformed outcomes, exp(coef) should be reported as % change – it’s not clear whether this is what is proposed

• Lines 218, 222: Please avoid the term ‘statistically significant’ and report all estimates with p-values, regardless of significance level.

Results

• Fig1: graphs are different sizes and dimensions

• Fig2: I can’t read some of these numbers- rethink colour scheme?

• Table S5: this table has very little information of use, and none of the % mentioned in the text are reported here.

• Table S6: what does the column ‘no guidance’ show?

Discussion

• Lines 356-269:.Can you expand on why they’re different eg what aspects of PA different metrics are capturing, when one might be more or less suitable than another?

• Line 375-378: a related point is that much of the evidence on which guidelines are based is still from self-report rather than accelerometer PA, and even in the latter case much of it is on older count-based approaches. Does this suggest we should try to use metrics that replicate this evidence even if they are not accurate? Or are the newer metrics presented here ‘better’ in any way? Can you say anything about when estimates of MBs might be over or underestimated?

• Line 388-391: what valuable insights?

• Line 391-397: what are the implications of this? Are you arguing that some metrics are better than others? Suitable for different purposes?

• Line 398-418: again, this is just reporting results – can you explain what this means for choosing different metrics?

Conclusion

• Line 435-437: can you offer a bit more guidance than just awareness? Surely reporting analyses using multiple metrics will just be confusing and encourage cherry-picking of results – what we need is guidance on which metric to use, and possibly when.

6. PLOS authors have the option to publish the peer review history of their article (what does this mean?). If published, this will include your full peer review and any attached files.

Reviewer #1: No

Reviewer #2: No

---

## [Author Response · Author response to Decision Letter 0]

18 Jul 2024

We would like to thank the editors and the reviewers for carefully reviewing our manuscript and the appreciation of our work. We believe the constructive comments and suggestions improved the quality of the manuscript. All changes are visible with track changes in the manuscript and are also integrated in this rebuttal letter/point-by-point reply. 

The following responses are also uploaded as a word document 'responses to reviewers" including a point-by-point reply. We recommend to read the uploaded word document as this contains a clearer overview of the the responses to the reviewers.

Reviewer #1: 

The results indicate that the movement behavior data varied depending on the metric used for analysis. Specifically, ENMO (Euclidean Norm Minus One) represented the most sedentary movement behavior profile, while CPM (Counts Per Minute) vector magnitude represented the most active profile.

This suggests that different accelerometer metrics capture different aspects of movement behavior, with some metrics highlighting more sedentary patterns while others emphasize more active behaviors. Understanding these differences is crucial for accurately assessing individuals' activity levels and sedentary behavior, which in turn can inform interventions aimed at promoting physical activity and reducing sedentary time to improve overall health outcomes.

Interestingly, the study found that reallocating time towards moderate-to-vigorous physical activity consistently predicted significant improvements in cardiometabolic variables, with the exception of fat percentage. This suggests that increasing time spent in moderate-to-vigorous physical activity may have positive effects on various aspects of cardiometabolic health, highlighting the importance of engaging in activities that elevate heart rate and promote greater exertion.

Overall, these findings emphasize the complexity of studying movement behaviors and their associations with health outcomes, indicating that the choice of accelerometer metric can influence both compliance rates with guidelines and the observed relationships with cardiometabolic variables.

Overall, the study suggests variations in agreement between different metrics, with some showing better consistency than others across various activity intensities. These findings underscore the importance of considering the choice of metric carefully when analyzing accelerometer data for physical activity assessment. Average acceleration showed strong associations with MVPA across all metrics, its relationship with LPA was weaker. Additionally, the intensity gradient exhibited stronger correlations with MVPA compared to LPA, with some variation across metrics.

The choice of metric significantly influences the composition of 24h MBs, particularly in terms of the distribution of time spent in sedentary behavior, light physical activity, and moderate to vigorous physical activity. ENMO tended to allocate more time to sedentary behavior, while CPM VM favored light physical activity, and MAD had a higher proportion of moderate to vigorous physical activity.

The analysis revealed significant associations between time spent on 24h movement behaviors (MB) and various cardiometabolic variables, with differences observed across different accelerometer metrics. Overall, these findings underscore the importance of considering different accelerometer metrics when examining associations between physical activity behaviors and cardiometabolic health outcomes. The direction and magnitude of associations varied across metrics, highlighting the need for tailored interventions aimed at promoting specific types of physical activity to improve cardiometabolic health.

The study has several limitations that should be considered when interpreting the results:

1. Selection of Cutoff Points: The use of cutoff points for each accelerometer metric is a potential limitation. While commonly used cutoff points were selected, there are alternative cutoff points available. These cutoff points are typically based on specific validation protocols conducted in particular populations. Therefore, the choice of cutoff points may impact the interpretation of physical activity data and comparisons across studies.

2. Limited Generalizability: The study only used hip-worn accelerometer data for the waking day. This may limit the generalizability of the findings, particularly when comparing them to studies using wrist-worn accelerometer data. Differences in acceleration patterns based on body location can influence the assessment of physical activity levels. Therefore, caution is needed when generalizing findings to populations or studies using different accelerometer placements.

3. Sample Characteristics: The results may be influenced by the characteristics of the sample population studied. Demographic factors such as age, gender, and physical fitness levels can affect physical activity patterns and associations with health outcomes. Therefore, the findings may not be representative of other populations with different demographic profiles.

4. Cross-Sectional Design: The study likely employed a cross-sectional design, which limits the ability to establish causal relationships between physical activity behaviors and cardiometabolic health outcomes. Longitudinal studies are needed to better understand the temporal relationships between these variables and to assess the effectiveness of interventions.

5. Measurement Error: Accelerometer measurements are subject to measurement error, which can arise from factors such as device malfunction, wear time compliance, and data processing methods. These errors could potentially affect the accuracy and reliability of the physical activity measurements and subsequent associations with health outcomes.

Acknowledging these limitations can help researchers and clinicians better interpret the study findings and guide future research efforts aimed at addressing these limitations to improve the understanding of physical activity and its impact on health.

Anwser: Thank you for the summary of the paper. We agree that the limitations mentioned above can improve interpretability of the study findings. Most of the limitations that were mentioned were already addressed in this paper. Therefore, we have added some sentences for additional clarification where necessary.

1. Selection of cut off points: The limitations linked to the selection of cut-points are already clearly mentioned in the discussion. However, we added an additional sentence to make this more clear.

Changes in manuscript

Discussion line 477-479: Although commonly used cut-points were selected, other cut-points are available, which are all based on a specific validation protocol in a specific sample. Using other cut-points might provide other results.

2. Limited generalizability: This limitation was already mentioned in the discussion line 485 – 487. “Finally, only hip-worn data for the waking day were used in this study, which hinders comparison with studies using wrist-worn data due to differences in acceleration based on body location.”

3. Sample characteristics: Since the aim of this study was to compare metrics on the same dataset, the sample characteristics are subordinate to the research question. It is our opinion that it is less important to mention the sample characteristics as a limitation. Therefore, we have chosen not to include this in the discussion/limitations section. 

4. Cross-sectional design: Associations between time-use and cardiovascular health parameters were moved to the Supporting Information S7_docx. Within the supporting information, we have added one sentence to highlight the inability to infer causal relationships. 

Changes in manuscript

Supporting Information S7_docx: The choice of data processing metric has an impact on the time spent in movement behavior features, average acceleration and intensity gradient. This supplementary information shows the impact of different 24h-MBs compositions, average acceleration and intensity gradient on the association with cardiometabolic health variables. However, because the data are cross-sectional causality cannot be inferred.

5. Measurement error: Devices with high calibration errors (n=1) were excluded. To ensure comparability, all datafiles were analysed with the same GGIR script, only deviating for the metrics used. See method line 147-149. “The GGIR package uses an autocalibration algorithm that checks and corrects for calibration errors in triaxial accelerometer signals [12]. Actigraph files (n=1) with a postcalibration error greater than 0.01 g were excluded [12].” This was also mentioned in the discussion line 474-476. “Perfect reproducibility across the four metrics was ensured by the use of the GGIR package, which allows for consistent data reduction features, i.e., autocalibration, sleep algorithm, and nonwear detection methods.”  

Reviewer #2: 

This paper focuses on the different metrics available in the GGIR package for accelerometer processing. This is quite a technical aspect of physical activity measures, but an important one, especially in light of the findings which show that the different metrics can produce quite different summary estimates. In general, I think this is a good, well-written, useful paper. However there are areas which are unclear and I feel it currently tries to tackle too much. The discussion would also benefit from less focus on just repeating the results and more on what this means for someone about to begin a study – which metric should they use and why? I list some more detailed points below.

Major points

1. Currently, definitions and advantages and disadvantages of the different processing metrics and PA summary measures don’t appear until the methods section (eg ENMO at line 142, intensity gradient line 160+), but this makes it very hard to understand the background and why these different metrics and summaries (and hence the manuscript itself) are important. As this manuscript involves very technical aspects of accelerometer processing which not all readers will be familiar with, I suggest adding a clear overview of the full process (raw data to processed data to cut points & summaries) early on in the background. Then describe the different processing metrics and the different PA measures - definitions, where they fit into this process, how they differ and how it might be expected to affect the summaries.

Answer: We agree and, as suggested, we now provide additional clarification in the background section. We now commence with a description of tri-axial accelerometers as the preferred method for analyzing 24-hour movement behaviors, highlighting the transition from processing 'counts per minute' data with closed-source software specific tools to processing raw accelerometer data with open-source tools. Next, we outline the derived outcomes from these data, including time spent in movement behaviors based on cut-points for classifying activity intensities, alongside newer features like cut-point-independent average acceleration and intensity gradient. We then proceed with the explanation of the concept of metrics, accompanied by cut-points for categorizing behaviors into activity intensities. Commonly used metrics such as ENMO, MAD, CPM VA, and CPM VM are highlighted, with the recognition that no gold standard exists for the selection of one metric. 

To enhance clarity, we have supplemented the background section with additional explanations in Table 1. This table now contains more detailed information (definition, formula and GGIR specifications) on the various metrics.

Changes in manuscript:

Background lines 42-90: To better understand 24h-MBs in adults, it is crucial to accurately measure these behaviors by measurement tools such as tri-axial accelerometers (e.g., the Actigraph GT3X+) [6]. These measurement tools are the preferred method for collecting 24h-MBs as they quantify accelerations in orthogonal directions of a three dimensional space [6,7]. There has been a shift from analyzing accelerometer data using "activity counts per minute" generated by closed-source proprietary accelerometer brand-specific algorithms (e.g. ActiLife software for Actigraph accelerometers) toward analyzing raw gravitational acceleration data (m/s²) [8]. Raw acceleration data allows for open-source data processing, such as the R package GGIR, which can be used regardless of the type of accelerometer [8,9]. Output from this open-source raw data package can be classified as time spent in movement behaviors defined by cut-points to classify activity intensities, as well as newer cut-point independent movement behavior features such as average acceleration and intensity distribution of activity throughout a day. These cut-point independent movement behavior features enhance the comparability between studies [10,11]. 

Nevertheless, working with raw data still requires the use of data reduction methods, also called metrics, to separate the acceleration signal from the gravitation signal [8,9]. Different metrics exist and these can be distinguished based on the method for extracting the acceleration signal [7,8]. Commonly used data reduction metrics for processing raw accelerometer data in GGIRare the Euclidian Norm Minus One (ENMO) and Mean Amplitude Deviation (MAD) as these analytic techniques are perceived as not too complex for users and they have the ability of quantifying output in universal units instead of abstract scales [see Supporting Information S1 for more details] [8,9,12]. As the shift from using activity counts cut-points to classify activity intensities into raw accelerometer data processing is still evolving, a new metric that replicates the closed-source Actilife software was developed in the GGIR package, i.e., the “counts per minute” (CPM) metric [13]. This CPM metric has the ability to process data of the vertical axis (VA) only or to work with the vector magnitude (VM) [See S1] [13]. The main advantage of this new metric in GGIR is the reduction of human errors. Data processing in ActiLife software requires manual processing of data to define wear and nonwear times, and the GGIR package applies the same nonwear algorithm on each data file [13]. Despite the popularity of working with accelerometer data, no gold standard exist for the most appropriate activity intensity-based cut-point accompanied by a data reduction metric. This lack of standardization affects the time spent in movement behavior and hampers comparability between studies [14]. 

Previous studies have highlighted that there are differences in cut-point dependent PA and SB durations depending on whether they are derived from raw accelerometer data or “counts per minute” data [14,15]. In contrast, literature shows comparable findings for cut-point independent average acceleration and intensity gradient across the acceleration metrics ENMO and MAD [16]. Interestingly, although 24h-MBs are codependent, none of these studies interpreting time spent engaged in behaviors used a compositional behavioral approach but focused on one or more behaviors in isolation (e.g., PA, SB). Moreover, no previous studies have compared three different movement behavior features (i.e., time spent in a 24h period, overall activity volume, and overall activity intensity) between the new CPM metric for VA and VM, the ENMO metric and the MAD metric with hip-worn accelerometer data in adults.

Supporting information S1_Table: See S1_table for adjustments.

2. This manuscript is trying to do a lot of things, which makes it difficult to follow. In particular, associations between PA measures and cardiovascular outcomes seem out of place – they’re not affected directly by the processing metrics, the analysis is not in-depth enough to be a full association study and its not clear what the implications are for different metrics. I think the paper would be stronger and clearer if this section was dropped entirely. If the authors do decide to keep this, then there needs to be more linking to the metrics and crucially some guidance on the appropriate metrics to use in different circumstances.

Answer: We agree, and have decided to move the analysis of associations between 24h compositions and cardi

---

## [Decision Letter · Decision Letter 1]

21 Aug 2024

A comparative analysis of 24-hour movement behaviors features using different accelerometer metrics in adults: implications for guideline compliance and associations with cardiometabolic health

PONE-D-24-08044R1

Dear Dr. Craemer,

We’re pleased to inform you that your manuscript has been judged scientifically suitable for publication and will be formally accepted for publication once it meets all outstanding technical requirements.

Kind regards,

Zulkarnain Jaafar

Academic Editor

PLOS ONE

Additional Editor Comments (optional):

Reviewers' comments:

Reviewer's Responses to Questions

**Comments to the Author**

1. If the authors have adequately addressed your comments raised in a previous round of review and you feel that this manuscript is now acceptable for publication, you may indicate that here to bypass the “Comments to the Author” section, enter your conflict of interest statement in the “Confidential to Editor” section, and submit your "Accept" recommendation.

Reviewer #2: All comments have been addressed

2. Is the manuscript technically sound, and do the data support the conclusions?

Reviewer #2: Yes

3. Has the statistical analysis been performed appropriately and rigorously? 

Reviewer #2: Yes

4. Have the authors made all data underlying the findings in their manuscript fully available?

Reviewer #2: Yes

5. Is the manuscript presented in an intelligible fashion and written in standard English?

Reviewer #2: Yes

6. Review Comments to the Author

Reviewer #2: I would like to congratulate the authors on a very thoughtful and comprehensive response to my comments. The extra explanation in the discussion and the moving of the association study to the supplementation file make the paper much clearer to read. I am satisfied with these and all the minor changes. While it’s a shame that they are not able to make any recommendations about the use of metrics in different situations, I understand the limitations here . One concern was that others might use this study to justify cherry-picking the metric that gives the most desirable answers, but I think that the caveats in the discussion should mostly discourage this.

This is really useful paper and I have no further comments.

7. PLOS authors have the option to publish the peer review history of their article (what does this mean?). If published, this will include your full peer review and any attached files.

Reviewer #2: No

---

## [Editor Report · Acceptance letter]

5 Sep 2024

PONE-D-24-08044R1 

PLOS ONE

Dear Dr. De Craemer, 

I'm pleased to inform you that your manuscript has been deemed suitable for publication in PLOS ONE. Congratulations! Your manuscript is now being handed over to our production team.

Kind regards, 

on behalf of

Dr. Zulkarnain Jaafar 

Academic Editor

PLOS ONE